# Marine Natural Products: Promising Candidates in the Modulation of Gut-Brain Axis towards Neuroprotection

**DOI:** 10.3390/md19030165

**Published:** 2021-03-19

**Authors:** Sajad Fakhri, Akram Yarmohammadi, Mostafa Yarmohammadi, Mohammad Hosein Farzaei, Javier Echeverria

**Affiliations:** 1Pharmaceutical Sciences Research Center, Health Institute, Kermanshah University of Medical Sciences, Kermanshah 6734667149, Iran; sajad.fakhri@kums.ac.ir; 2Student Research Committee, Faculty of Pharmacy, Kermanshah University of Medical Sciences, Kermanshah 6714415153, Iran; aki_yar@yahoo.com (A.Y.); Mostafa_yar@yahoo.com (M.Y.); 3Medical Technology Research Center, Health Technology Institute, Kermanshah University of Medical Sciences, Kermanshah 6734667149, Iran; 4Departamento de Ciencias del Ambiente, Facultad de Química y Biología, Universidad de Santiago de Chile, Santiago 9170022, Chile

**Keywords:** marine natural products, gut-brain axis, neuroprotection, signaling pathway, therapeutic target, pharmacology

## Abstract

In recent decades, several neuroprotective agents have been provided in combating neuronal dysfunctions; however, no effective treatment has been found towards the complete eradication of neurodegenerative diseases. From the pathophysiological point of view, growing studies are indicating a bidirectional relationship between gut and brain termed gut-brain axis in the context of health/disease. Revealing the gut-brain axis has survived new hopes in the prevention, management, and treatment of neurodegenerative diseases. Accordingly, introducing novel alternative therapies in regulating the gut-brain axis seems to be an emerging concept to pave the road in fighting neurodegenerative diseases. Growing studies have developed marine-derived natural products as hopeful candidates in a simultaneous targeting of gut-brain dysregulated mediators towards neuroprotection. Of marine natural products, carotenoids (e.g., fucoxanthin, and astaxanthin), phytosterols (e.g., fucosterol), polysaccharides (e.g., fucoidan, chitosan, alginate, and laminarin), macrolactins (e.g., macrolactin A), diterpenes (e.g., lobocrasol, excavatolide B, and crassumol E) and sesquiterpenes (e.g., zonarol) have shown to be promising candidates in modulating gut-brain axis. The aforementioned marine natural products are potential regulators of inflammatory, apoptotic, and oxidative stress mediators towards a bidirectional regulation of the gut-brain axis. The present study aims at describing the gut-brain axis, the importance of gut microbiota in neurological diseases, as well as the modulatory role of marine natural products towards neuroprotection.

## 1. Introduction

The Modern lifestyle with the consumption of processed foods, meat, and wheat has changed the normal flora of the gastrointestinal tract (GIT) [1]. Recent studies are triggering the idea of revealing the relationship between gut flora and central nervous system (CNS) disorders like Parkinson’s disease (PD), Alzheimer’s disease (AD), multiple sclerosis (MS), amyotrophic lateral sclerosis (ALS), autism spectrum disorders (ASD), and mood disturbances such as anxiety and depression [2]. Growing evidence has proved the bidirectional communication of GIT and CNS named gut-brain axis. Regulation of the gut-brain human physiology may be affected by billions of bacteria that reside in the body. GIT is the main place that keeps the majority of this flora and these residents are called gut microbial (GM) [3]. Gut homeostasis can be compromised by several factors, including antibiotic exposure, diet, and infections and this alteration in the composition of GM takes part in the pathogenesis of gut-brain-associated diseases [4]. The Gut-brain axis is the main complex anatomical way in which the gut and brain hold their bidirectional relationship and could communicate with each other in health and diseases. Studies have shown the impact of GM on brain evolution, mood, and immune function [3]. GM communicates with gut epithelium to improve the body’s hemostasis and immunity. The strongest evidence for the role of GM on brain development was obtained from studies on germ-free (GF) mice [2]. In this regard, the dysregulated composition of gut bacterial plays a crucial role in the pathogenesis of gut-brain disorders [5]. This shows a bidirectional relationship through which the disturbance in the GM might influence the neurological signs and vice versa [2]. In another word, with the conception of a gut-brain axis being put forward, there is an increasing belief that this communication acts bidirectionally through which GM influence CNS, and the CNS affects the GM. Growing studies suggest that GM affects the development, functions, and disorders of the central nervous system through the regulation of associated receptors and signaling mediators [6]. The neuroimmune and neuroendocrine systems are two critical compositions of the gut-brain axis [7]. Revealing the detailed microbiota functions mediated by pivotal dysregulated pathways is essential to our finding of how the gut-brain axis may influence the neuronal outcomes [8]. Besides, dysregulation of intestinal permeability and gut integrity affects gut-bacteria-derived metabolites and related signaling pathways, towards the progression/development of different neurological diseases [9]. So, the GM helps to restore the normal function of the nervous system and gut-brain signaling. Several molecular mechanisms are behind the gut-brain bidirectional relationship. Regarding revealing the molecular insights into the influence of GM on CNS, it has been shown that gut-microbiota communicates CNS through producing multiple metabolites/neurotransmitters with neuromodulatory properties. Of those, inflammation, apoptosis, and oxidative stress, as well as associated signaling pathways/mediators play critical roles in facilitating the bidirectional relationship of gut and brain. Accordingly, gamma-aminobutyric acid (GABA), glutamine, 5-hydroxytryptamine (5-HT), histamine, glial cell function, synaptic pruning, blood-brain barrier function (BBB), and myelination are important players [6,10]. Considering the gut-brain axis, migration of toxic agents from gut to brain trigger astrocyte activation via the activation of phosphoinositide 3-kinase (PI3K)/protein kinase B (Akt)/mammalian target of rapamycin (mTOR) pathway [11]. Instead, during pathological conditions, the aforementioned pathways/mediators tend to be involved in many devastating neurological situations.

There are also several pathophysiological mechanisms behind neurodegeneration, including oxidative stress, neuroinflammation, apoptosis, imbalances of calcium ions, malfunction of mitochondria, impairment of signal transport through axons, DNA damage, and abnormalities in RNA processing [12,13]. Accordingly, modulation of these factors seems to pave the road in the prevention/treatment of neuronal-associated disorders. Without knowledge about the precise mechanism and etiology lying behind these disorders, they have some recurrent traits such as malfunction of mitochondria, protein misfolding, and inadequate clearance, which make them complicated to deal with. Complex pathological pathways of neurodegenerative diseases ensure the need of using natural molecules with diverse pharmacological properties [14]. By having 70% of plants covering the earth, and diverse organisms living, the marine environment is the most significant source for natural products. Rich biological and genetic diversity is owed to the harsh environmental conditions of the oceans. One of the preferences of natural medicine over synthetic is their better tolerance. It has been also shown that marine natural products have antioxidative, immunomodulatory, and anti-inflammatory properties [15].

Marine natural products such as carotenoids, polysaccharides, phytosterols, terpenoids, macrolactins, and alkaloids apply potential antioxidant and scavenging characteristics in modulating the properties of the gut-brain axis. Recent reports have shown the association of GM and neurodegenerative diseases [2,11,16] through those inflammatory/apoptotic/oxidative stress pathways. To the best of our knowledge, this is the first review highlighting the potential of marine-derived natural products in modulating the gut-brain axis towards neuroprotection. In this work, potential roles of marine-derived natural products have been reviewed on the gut-brain axis with respect to neurodegenerative diseases. Additionally, the association between gut microbial composition and CNS in physiological and pathological conditions has been described. Also, the applicable rationale for using marine-derived natural products in treating and managing neurodegenerative diseases has been reviewed. Marine natural products could be introduced as alternative candidates in the modulation of the gut-brain axis towards neuroprotection.

## 2. Gut Microbiome and Gut-Brain Axis in Diseases

Regulation of human physiology may be affected by billions of bacteria that reside in the body. It was estimated that there are 10^11^ bacteria per gram of the colon contents [17,18]. They are not the sole residents of this ecosystem, and viruses, protozoa, archaea, and fungi are also present [19]; however, the microbiota seems to be the king of GIT. The GIT is the main place that keeps the majority of this flora and these residents are called GM. Four major and two minor groups of GM are *Bacteroidetes*, *Firmicutes*, *Proteobacteria*, *Actinobacteria,* along with *Verrucomicrobia, Fusobacteria,* respectively [20].

Early investigations about interactions between GIT and the brain were focused on digestion and satiety [21]. Homeostasis of the gut is maintained by interactions of these bacteria with each other and with the epithelium layer of the gut and this homeostasis leads to improvement of host immunity [22,23]. The composition of GM is affected by so many factors and the hosted nature plays a key role on these inhabitants. Other factors include genetic, age, physical activity, environmental factors, infection, exposure to antibiotics [24], altering mucus secretion-induced stress [25], and nutrition [5,26]. In recent decades, nutrition and foods have attained special attention towards modulating GM.

The presence of GM has a double role in health and disease states, improving immune functionality and progression/suppression of the diseases, including neurodegeneration [22,23], such termed gut-brain axis. The gut-brain axis is the main complex anatomical way in gut and brain hold their bidirectional relationship and could communicate with each other in health and diseases [27]. The complexity of the relationship between GM and the CNS was exhibited in relevant studies. However, the exact mechanism of the bidirectional gut-brain relationship is not known. Studies showed that GF mice couldn’t develop a healthy intestinal tract in comparison to specific pathogen-free (SPF) and GF-conventional mice, confirming this hypothesis which GM has an essential function in the development of the enteric nervous system (ENS) [6,28], CNS, and hypothalamic-pituitary-adrenal (HPA) axis [29] in early stages of the postnatal life.

In this line, beneficial changing of GM via using antibiotics in SPF mice resulted in elevated pro-survival brain-derived neurotrophic factor (BDNF) in the hippocampus and depleted expression in the amygdala [30]. The Gut-brain axis consists of CNS (brain), ENS, and the digestive system [24]. Intrinsic innervation of the gut is achieved by the complex network of ENS neurons, including two networks, the myenteric and submucosal networks, that modulate gut functions like peristalsis, secretion, and absorption [31]. It is involved in peristalsis, hormone/acid secretion, bicarbonate, and mucus production [24]. The vagus nerve is the major pace to transmit visceral signals into CNS to cause reflexes and mind/moods change and the brain signals to gut for modulation of gut’s physiology and function [6,27].

The autonomic nervous system (ANS) in association with neuronal and neurohormonal signaling, controls many of the physiological functions such as breathing, heartbeat, digestion, peristalsis, bile secretion, permeability, carbohydrate level, mucosa state, homeostasis of mucosal fluids and osmolality, mucus production and mucosal immune functions [27,32]. ANS synapses are the places that sense the microbial metabolites as the tools of communication of microbiota with each other [33]. ANS directly signals gut via CNS leading to changes in its physiology. Gut epithelium’s role in the activation of immune responses can be modulated by ANS in both direct and indirect ways. In direct ways, it modulates the response of gut immune cells to microbes, and in indirect ways, it modulates microbes [34].

The microbiota-gut-brain (MGB) axis makes a bidirectional relationship between microbiota and brain [27], through which any disturbance in one of their functions, like the composition of the gut microbial, will affect the other one [2]. Any disruption in gut-brain crosstalk could result in the progression of cognitive and neural diseases [35,36]. The MGB axis is a component of a huge physiological network complex, including the endocrine (HPA axis), immune system (mediated by cytokines and chemokines), ANS, and the ENS. Gut microbiota exerts its effect on the HPA axis and vagus nerve by metabolites produced from the metabolism of tryptophan [24]. In this line, several microbial molecules take part in MGB-associated signaling and communication [37,38]. It has been revealed that *Clostridium sporogenes* decarboxylases could turn tryptophan to tryptamine, a neurotransmitter that causes serotonin and dopamine released by neurons [39]. Besides, one of the important inhibitory neurotransmitters like GABA is produced by *Lactobacillus* spp. and *Bifidobacteria* spp. from glutamate [40]. Cells that participate in these signaling pathways in association with bacterially derived compounds are enteroendocrine cells (EECs), enterochromaffin cells (ECCs), and mucosal immune cells. Stimulated EEC cells lead to the production of neuropeptides like, peptide YY, neuropeptide Y (NPY), and substance P that affect ENS [41]. To provide the precise molecular aspects of the effects of the GM on CNS, it has been shown that GM communicates CNS directionally through producing multiple neurotransmitters/metabolites with neuromodulatory properties. Amongst the mediators/pathways, glutamine, histamine, synaptic pruning, glial cell function, BBB function, and myelination are important players [6,10]. In addition to the above-mentioned methods of communication, the microbiota can directly synthesis the neuroactive mediators like GABA [42], 5-HT, norepinephrine, and dopamine [43]. Considering the gut-brain axis, the migration of toxic agents from gut to brain could trigger cell migration in astrocytes via the activation of the PI3K/Akt/mTOR pathway [11]. Consequently, the aforementioned mediators have critical roles in several body procedures, including apoptosis, inflammation, oxidative stress, as well as cell migration and proliferation towards homeostasis and even pathological situations of different organs.

The upstream factors such as growth receptors, G protein-coupled receptors (GPCRs), receptor tyrosine kinases (RTKs), and cytokines play a critical role in the attenuation of Janus kinase (JAK)/signal transducer and activator of transcription (STAT). Furthermore, Akt phosphorylates glycogen synthase kinase 3 (GSK-3β) with critical roles in several disorders especially neurodegenerative diseases. Akt also influences apoptotic pathways (e.g., Bax/Bcl-2, caspases), inflammatory mediators (Ils, COX, NF-κB), and oxidative factors (e.g., SOD, ROS, Nrf2, HO-1, CAT) in combating diseases/neurodegenerative disorders [44]. In this line, treatment with probiotics reduced depressive-like behaviors through increasing Bcl-2 and p-Akt, while decreasing malondialdehyde (MDA), cleaved caspase-3, and Bax in the serum [8].

As provided, the aforementioned GM plays a major role in the metabolization of natural compounds towards biological activities and health benefits, especially in neurodegenerative diseases. In this line, actively produced compounds can attenuate signaling mediators involved in neurodegeneration. Although, some of these metabolites cross the intestinal barrier and reach BBB via the bloodstream [38,45].

## 3. Gut-Brain Axis in Neurodegenerative Disease

Nowadays, neurodegenerative diseases have been considered global concerns and a lot of studies have focused on this area of research to lower burdens of associated disorders [46]. Neural diseases such as PD and AD consist of a group of disorders that 1% and 8% of the population suffer from CNS and peripheral nervous system (PNS) deteriorations, respectively [47].

Recent studies have shown that altered normal GM has a crucial role in neurodegenerative diseases like PD, AD, ALS, and depression; however, the exact mechanism underlying this phenomenon needs to be more scrutinized [48]. A recent study confirmed the relationship of microbiota dysbiosis (i.e., alteration of GM) and AD pathology [49], decreased count of gram-negative species and increased intestinal permeability also have been noted [48]. In AD, the reduction of GM biodiversity by taking antibiotics resulted in changes of neuroinflammatory and amyloidosis that confirmed the role of GM in the pathology of AD [50]. One of the key contributors to the pathogenesis of AD is microbiota diversity. For example, it was seen that in developed countries with a high hygienic condition, lower diversity of GM is correlated with AD incidence [16]. During AD, some changes have been seen in GM, including *Bacteroides vulgatus*, *Bacteroides fragilis*, *Eggerthella lenta*, *Odoribacter splanchnicus*, *Butyrivibrio hungatei*, *Butyrivibrio proteoclasticus*, *Eubacterium eligens*, *Eubacterium hallii*, *Eubacterium rectale*, *Clostridium sp.*, *Roseburia hominis*, *Bifidobacterium bifidum*, and *Faecalibacterium prausnitzii*. This leads to increased accumulation of cerebral amyloid β (Aβ)/neuroinflammation, increasing bacterial lipopolysaccharides (LPS), as well as elevated interleukin (IL)-1 beta, NLR family pyrin domain containing 3 (NLRP3), and chemokine (C-X-C motif) ligand 2 (CXCL2). Dysregulated levels of toll-like receptors (TLRs), nuclear factor κB (NF-κB), IL-1β, IL-18, Aβ, and caspase-1 also results from the bidirectional dysregulation of the gut-brain axis in AD [51,52,53,54].

The role of human microbiota is more explored by advanced technology in the last two decades in both physiological and pathological states. Up to date methods of GF animal models, antibiotic aided microbiota manipulation and fecal microbial transplantation have been used in investigations [12]. For example, in an investigation to find the role of GM in the prevalence of AD in elder patients, they compared the fecal samples of AD patients with healthy elders. They found that a lower prevalence of butyrate-producing bacteria along with a higher abundance of bacterial taxa could be used as predictors of AD [51]. Some bacteria species such as *Firmicutes*, *Bacteroidetes*, and *Proteobacteria* are involved in the pathogenesis of chronic inflammatory diseases through produced amyloids including IL-17A and IL-22 cytokines [55]. The role of GM in the production of vitamin B12, which has a key role in cognitive abilities, is another example that emphasizes its importance [56]. In an investigation to find the role of GM in the prevalence of AD in elder patients, researchers compared the fecal samples of AD patients with healthy elders. They found that lower counts of butyrate-producing bacteria with a higher abundance of bacterial taxa could be used as predictors of AD [51].

One of the main barriers to restoring the GM is age and associated diseases. As the age of the patient goes up, the restoration will be more compromising. In patients with PD, the alteration of the GM and infection with *H. pylori* has been noticed [2]. In PD, dysregulated levels of the GM, *Enterobacteriaceae, Prevotellaceae, Verrucomicrobiaceae, Lactobacillus, Porphyromonas, Parabacteroides, Mucispirillum,* and *Bacteroides fragilis* results in an elevated rate of TLR4, IL-1β, IL-2, IL-4, IL-6, IL-13, IL-18, tumor necrosis factor-α (TNF-α), and interferon (IFN)-γ [3]. The PD metagenome includes more levels of genes taking part in LPS biosynthesis and type 3 bacterial secretion systems which show the higher potential of inflammation by microbial metabolites [57]. These studies reinforce the role of circulating inflammatory products in the periphery CNS inflammation presented in PD [50]. Toxic alpha-synuclein (α-Syn) aggregates are the hallmark of the Lewy bodies that are well known as the marker of PD substantia nigra pars compacta neurons [58]. In a study, it was shown that the first site of α-Syn deposition was the submucosal layer of the intestine [59]. In an analysis done on the fecal samples of PD patients, higher amounts of *Enterobacteriaceae*, and lower counts of *Prevotellaceae* were obvious in comparison to the control group with the same age. Raised levels of *Enterobacteriaceae*, as well as depleted amounts of *Prevotellaceae*, showed a correlation with postural and walking disabilities. The effect of *Prevotellaceae* is due to its ability in generating short-chain fatty acids (SCFAs)*,* thiamine, and folate as by-products to create a healthy environment [60].

Probiotics are defined as living microorganisms, that have beneficial effects on consumer health when enough amounts of them have been digested. Their uses have increased in medical and clinical fields with altering impacts on other CNS disturbances, including anxiety and depression [61,62]. Their effect on GM proliferation has been proved [24]. In patients with anxiety/depression *Bifidobacterium*, *Alistipes*, *Prevotella*, *Parabacteroides*, *Lachnospiraceae*, *Anaerostipes*, *Oscillibacter*, *Faecalibacterium*, *Ruminococcus*, *Clostridium*, *Megamonas*, *Streptococcus*, *Klebsiella*, and *Phascolarctobacterium* are changing towards decreasing dopamine (DOPAC), homovanillic acid, hippocampus 5-HT, BDNF expression, and circulatory IL-10 while increasing plasma stress hormone [63,64]. Additionally, dysregulated levels of GABA, dopamine, 5-HT, and IL-10 have also been shown in anxiety/depression associated with the gut-brain axis [63,64,65]. Stress-related mental disorders like anxiety and irritable bowel syndrome (IBS) are highly correlated. This correlation triggered the idea of gut-brain axis studies. More than 50% of patients suffering from IBS have comorbidities of anxiety and depression [66]. In a study conducted by Sudo et al., it was shown that undisturbed GM composition at the early stages of life has a huge impact on adulthood stress management [29]. Later research showed that this matter will affect neurochemical compounds like, cortical and hippocampal brain-derived neurotrophic factor [29,67], hippocampal 5-HT receptor 1A magnitude [67], striatal monoamine turnover [68], and gene expression of synaptic plasticity [68] which emphasizes the potent influence of GM on CNS phenotype. Moreover, other effects of GM are in anxiety [67,68] and depression [69], pain response [70], feeding, taste, and metabolism areas [71].

In addition to AD, PD, anxiety/depression pain, and aging, there are other neurological deteriorations (e.g., ALS) that are usually associated with altered GM and lowered biodiversity of the gut flora suggesting the interrelationships of these factors. Neuroimmune activation may be achieved by increasing the levels of butyrate-producing species [48]. Also, in stool samples of ALS patients, higher levels of inflammatory *Ruminococcaceae*, *Enterobacteria*, and *Escherichia coli* have been detected in comparison to the control group [72]. Pre-clinical results showed increased gut permeability, damaged tight junction structure, and increased numbers of abnormal Paneth cells, a cell type responsible for antimicrobial defense in animal models of ALS. Besides, GM indicated a shifted relative abundance of microbial species including a decrease in butyrate-producing *Butyrivibrio fibrisolvens* [73]. Clinical evidence also confirmed a meaningful increase in *Bacterioidetes* and, consequently, a decreased *Firmicutes* to *Bacteroidetes* level, as well as a decrease in the beneficial *Anaerostipes*, *Lachnospiraceae*, and *Oscillobacter* in the GM of ALS patients. The aforementioned functional changes in ALS patients were concluded to be associated with the dysregulated levels of nitric oxide (NO), GABA, LPS, AMPA/*N*-methyl-D-aspartate (NMDA), and oxidative pathways [74,75,76].

Communication of gut and brain is mediated by some bacterial products towards neurological signs. In MS disease, a study by Farrokhi and co-workers showed lowered serum level of lipid 654 as the metabolite of *Bactroidetes* spp. In comparison to the control group [77]. In another study, it was demonstrated that *Clostridium perfringens* toxins B and D [78] could cause MS like-symptoms, including blurred vision and motor function disability [56]. Toxins-induced visual defects in MS patients are due to retina inflammation caused by formed defects in barrier veins and binding to vascular receptors [79]. Patients with MS experience the changes in levels of *Acinetobacteria*, *Bacteroidetes*, *Desulfovibrionaceae*, *Firmicutes*, *Proteobacteria*, *Verrucomicrobia*, and associated genus [77,80]. This is in agreement with dysregulated GABA, reduced levels of 5-HT and dopamine, while increased IFN-γ, monocyte chemoattractant protein (MCP-1), macrophage inflammatory protein (MIP)-1α, MIP-1β, and IL-6 in MS patients [77,80,81,82].

As another neurological disorder that has an undeniable interconnection with GM, ASD has several changes in gut bacteria, including *Bifidobacteraceae*, *Veillonellaceae*, *Lactobacillaceae*, *Bifidobacterium*, *Megasphaera*, *Mitsuokella*, *Rumnicoccus*, *Lachnoclostridium*, *Clostridium*, *Sutterella*, *Desulfovibrio*, *Lactobacillus*, *Eubacterium*, *and Prevotella* [83]. These changes are concomitant with pathophysiological changes in signaling mediators, including the upregulation of mTOR, TNF-α, IL-4, IL-5, IL-6, IL-8 while down-regulating IL-10, transforming growth factor beta (TGF-β), and 5-HT in ASD [11,81,84,85,86,87].

From the mechanistic point of view, behind the gut-brain association, oxidative stress and inflammation seem to play more important roles. Oxidative stress is one of the significant factors involved in mitochondrial dysfunction that has been observed in neurodegenerative diseases. It is the result of an imbalance between generated reactive oxygen species (ROS) and antioxidant defense arsenal. Biological targets for ROS molecules are lipids, proteins, and nucleic acids that lead to their destruction and degradation [88]. It has been seen that communication of microbiota with host cells can be done by merging them with mitochondrial activities. Potentially, interactions of microbiota-gut-brain axis with CNS oxidative stress may exist. In this line, increased amounts of ROS are associated with dysbiosis of microbiota leading to inflammation of the CNS. On the other hand, malfunction of CNS caused by brain lesions could lead to alterations in GM composition. Relationship between oxidative stress-mitochondria-microbiota and neurodegenerative diseases accents the importance of the gut-brain axis [18]. It has been shown that stress has an impact on postprandial gastrointestinal motility and induces a temporary reduction of gastric emptying in dogs [25]. Stress applies its effect through stress mediators, causing local immune activity via alteration of intestinal permeability [89] and can induce changes in germ composition [90].

Several studies have also shown the impact of GM on the CNS and immunity system. However leaky gut syndrome (LGS), which is the permeation of the normal flora into the outside of the intestinal lumen and consequently increased levels of neuroactive metabolites causes a neuroinflammatory response in the brain including cerebellum and hippocampus dysfunction [91,92]. It was evidenced that LGS is common in patients with multiple CNS disorders [93] and leaked metabolites into the blood compromises CNS [94]. Chronic mild inflammation leads to the release of cytokines into the blood and affecting the immune system. Inflammation-induced by microbiota is mediated by molecules such as LPS and peptidoglycans. Recognition of LPS is done by TLR4 which monocytes, macrophages, and brain microglia are rich in them [24]. Studies showed the presence of TLR4 mediated inflammatory responses in depressed IBS patients [95,96]. Blood levels of pro-inflammatory and anti-inflammatory chemokines can be modulated indirectly by microbiota and probiotics that have a direct effect on brain functions [24]. By introducing *E. coli* to the GF mice, macrophage activation and infiltration in adipose tissue led to high levels of pro-inflammatory cytokine and IFN expression [97].

From another mechanistic point of view, the GM has been shown to alter the evolution, activities, and abnormalities of the CNS and ENS by binding and stimulating pattern recognition receptors (PRRs) such as TLR2 and TLR4 [6,98]. Imbalance of the GM community, distortion of gut integrity and permeability leads to increasing levels of microbial products and microbes associated molecular patterns (MAMPs) in mesenteric lymphoid tissues that cause the occurrence of different neurological diseases [2,9]. Comparison of GF animals with conventional control mice confirmed that hormone signaling, BDNF expression, neurotransmission, and amino acid metabolism, was impaired in GF models [99]. The changes in microbiota composition due to taking antibiotics, influence the integrity and activities of the ENS, neurochemistry, and decrease the number of ganglia residing enteric glial cells in vivo [100].

The relatively fixed composition of the GM throughout life will be compromised in a series of situations such as, illnesses, exposure to antibiotics, and changing of diet or lifestyle [2]. Depending on the severity of the situation that the person encounters, the flora restores to the previous normal flora quickly or with a delay. But in some instances, it never goes back and turns to a chronic issue.

Altogether, GM and neurodegenerative diseases are in a bidirectional relationship, and modulating each of the aforementioned systems could affect the other. Table 1 shows changes in GM during some neurodegenerative diseases and related pathophysiological outcomes.

## 4. Marine-Derived Natural Products and Associated Sources

By covering 70% of planet earth and store a huge diversity of organisms, the ma-rine environment has been an outstanding source of natural products [88]. Associated genetic diversity and biological activities of marine active ingredients are owed to harsh environmental conditions of the oceans, living in conditions of high pressure, cold temperatures, dark fields, and adaptation to stressful conditions [101]. The molecular weight of marine-derived natural products ranges from 100 to 1000 Da and is specific to an individual taxonomic classification [102]. The survival of GIT microorganisms, which compete with other microorganisms, is greatly dependent on the produced or external administered marine-derived natural products. They take part in attraction, repletion, and even killing other competitors. Both eukaryotic and prokaryotic microorganisms can produce secondary metabolites. For example, *Bacillus* spp., *Pseudomonas* spp., eukaryotic fungi (e.g., *Penicillium* spp., *Aspergillus* spp.), filamentous actinomyces (e.g., *Streptomyces* spp.), and terrestrial plants have shown to produce secondary metabolites or associated metabolites.

Recently by discovering new secondary metabolites that are biologically active, one of the goals of the pharmaceutical and agrochemical industries is to produce them on large scale. Based on the diversity of the structures of secondary metabolites, they have huge potential to be used against diverse ranges of illnesses [103]. One of the features of marine natural sources is that they can act as a continuous source of bioactive molecules [88]. Two major sources of marine-derived natural products are marine organisms and fungi [104]. Investigations showed that the production of these metabolites is not a random phenomenon and is related to the ecological niche [105]. According to this fact, chemists who working on marine natural products, are trying to discover new species which produce these metabolites [104] through unique metabolic and genetic pathways [88]. So, the marine organisms and fungi absorbed a lot of attention [106]. Until now, more than 100 metabolites could be found that are produced by marine fungi [106].

Many marine-derived natural products have been found to elicit a broad range of bioactivities and, therefore, continue to be a prolific source for the production of new drugs or drug leads. We believe that the discovery of new and extreme habitats will advance the discovery of novel macro- and microorganisms and, thus, might lead to the detection and isolation of novel marine-associated natural products [107]. Of microorganism source of marine-derived natural products, Eubacteria are of major ones, including *Actinobacteria, Cyanobacteria*, and other bacteria. Besides, *Archaebacteria,* Euglenoids (Euglenozoa, Euglenoidea, Protozoa), Dinoflagellates, (Dinozoa, Dinoflagellatea, Protozoa), Ciliates (Protozoa, Ciliophora), Chrysophytes (Phaeophyta, Chrysophyceae, Chromista), Diatoms (Diatomae, Bacillariophyceae, Chromista), Eustigmatophytes (Phaeophyta, Eustigmatophyceae, Chromista), Raphidophytes (Chromista, Raphidophyta), Prymnesiophytes (Prymnesiophyta, Chromista), Cryptophytes (Chromista, Cryptophyceae, Cryptophyta), Prasinophytes or grass-green scaly algae (Plantae, Prasinophyta), Green microalgae (Chlorophyta, Plantae), Red microalgae (Rhodophyta, Plantae), and Fungi (Eumycota) [108]. Specifically, marine fungi are still an underestimated but rich source for new secondary metabolites, although their distribution and ecological role often remain scarce. Marine fungi are the sources of biologically active molecules with the known anticancer, neuroprotective, anti-angiogenesis, antibiotic, antiviral, antioxidative, and anti-inflammatory activities [109].

## 5. Marine-Derived Natural Products against Diseases: Approaches to the Gut-Brain Axis

As provided, marine-derived natural products could be extracted from several sources, especially bacteria, fungi, microalgae. Algae with a common name of seaweed are one of the major sources of marine-based compounds which are widely used in industries [110]. Some of those most important marine metabolites are carotenoids, polysaccharides, phytosterols, terpenoids, phenolic compounds, and alkaloids [111]. Based on their antioxidative, anti-inflammatory, and immune-regulatory characteristics, the aforementioned compounds showed satisfying results in the management of patients with diabetes, obesity, brain trauma, ischemic stroke, and other neurodegenerative diseases [112]. Neurodegenerative diseases are the result of physiological and pathological changes like ischemic strokes and brain injuries which end up in the loss of some neurons in specific regions of the brain [113]. Without knowledge about the precise mechanism and etiology lying behind these disorders, they all have features such as oxidative stress, neuroinflammation, malfunction of mitochondria, protein misfolding, and inadequate clearance, which make them complicated to deal with [114]. One of the preferences of natural medicine over synthetic is their better tolerance. It has been shown that marine natural products have antioxidative, immunomodulatory, and anti-inflammatory properties [115]. Complex pathological pathways of neurodegenerative diseases, ensure the need of using marine natural molecules with diverse pharmacological properties [116].

### 5.1. Carotenoids: Fucoxanthin, Astaxanthin, and Lycopene

As marine-derived compounds, carotenoids are fat-soluble pigments in plants, algae, fungi, and photosynthetic bacteria. These pigments generate bright yellow, red, and orange colors in plants, vegetables, and fruits. There are more than 600 different kinds of carotenoids with antioxidant effects [117]. Their relationship with photosynthesis is grouped into two classes, one group is directly involved in photosynthesis and another group protects the organism from photooxidation [118]. Major carotenoids produced by marine microorganisms include astaxanthin, fucoxanthin, lycopene, salinixanthin, saproxanthin, sioxanthin, siphonaxanthin, canthaxanthin, β-cryptoxanthin, diadinoxanthin, dinoxanthin, echinenone, lutein, zeaxanthin, and violaxanthin [23].

Compounds obtained from marine algae have potential antioxidative properties. Xanthones are marine natural products containing a tricyclic symmetric structure derived from dibenzo-γ-pirone [119]. Around 200 xanthone molecules have been recognized with the sources of plants, lichens, bacteria, and fungi [120]. Their biological activities include diverse ranges of antioxidative [121], antiproliferative [122], antimicrobial [123], antitumoral activities [124], and this diversity is due to their interactions with multiple molecular targets [125].

One of the most important xanthones is fucoxanthin with promising effects. Fucoxanthin is a carotenoid with several biological activities and health benefits through exerting anti-inflammatory effects in vitro and in vivo [126,127]. It has also been shown that fucoxanthin suppressed the cell cycle and induced apoptosis in combating cancer [128]. The hepatoprotective, cardioprotective, and anti-diabetic effects of fucoxanthin, as well as its effect on metabolic syndrome [129], are also indicated in concomitant studies [130]. Fucoxanthin is a product of *Sargassum siliquastrum* (a brown algae) and protects DNA from oxidation [16]. Brown algae, as an origin of fucoxanthin, showed antioxidant and anti-inflammatory effects in glial cells [131]. However, several other brown algae are marine sources for fucoxanthin, including *Sargassum siliquastrum*, *Hijikia fusiformis*, *Undaria pinnatifida*, *Laminaria japonica*, *Alaria crassifolia*, and *Cladosiphon okamuranus* [117]. Results of research recommend using algal metabolites, especially fucoxanthin in CNS diseases [132,133]. Fucoxanthin depleted the generation of Aβ1–42 fibril and Aβ1–42 oligomers, when co-incubated with Aβ1–42 monomers and showed the inhibitory effect of Aβ aggregation [134]. Besides, fucoxanthin prevents damages of DNA via H_2_O_2_ which is accompanied by increased levels of glutathione (GSH) and superoxide dis-mutase (SOD) [135]. It also protects LPS-activated BV-2 microglia via nuclear factor erythroid 2-related factor 2 (Nrf2)/heme oxygenase (HO)-1 pathway and promotes survival of the cell through the cAMP-dependent protein kinase (PKA)/cAMP response element-binding (CREB) pathway and increasing BDNF secretion [136]. Fucoxanthin also protects Aβ42-induced BV2 cells from inflammation via the reduction of pro-inflammatory mediators such as TNF-α, IL-6, IL-1β, and prostaglandin (PG)E2. Expression of inducible nitric oxide synthase (iNOS) and cyclooxygenase-2 (COX-2) and phosphorylation of mitogen-activated protein kinase (MAPK) pathway was reduced under influence of fucoxanthin [135]. Reduced expression of iNOS and COX-2 and secretion of inflammatory factors such as TNF-α, IL-6, PGE2, and NO take part in inhibition of Akt/NF-κB and MAPKs/stimulating protein-1 (AP-1) pathways, in LPS-activated BV-2 microglia was observed as the protectant activity of the fucoxanthin [135]. One of the major contributors to the pathologic processes of AD is the deposition of Aβ [137,138].

Oligomers of Aβ are notorious for their neurotoxicity and are one of the key compounds which take part in neurodegeneration of the AD. It is also demonstrated that fucoxanthin with its antioxidative and anti-apoptotic properties could play a protective role against oligomers of Aβ in SH-SY5Y cells. PI3K/Akt cascade as a protectant mechanism will be disturbed by oligomers of Aβ and the destructive series of reactions governed by the extracellular signal-regulated kinase (ERK) pathway will be activated. It has been also shown that inhibition of GSK-3β and mitogen-activated protein kinase (MEK) together could stop the destructive effects of Aβ. So, it could be concluded that PI3K/Akt and ERK pathways may contribute to Aβ oligomer-stimulated neurotoxicity. Effects of Aβ on PI3K/Akt and ERK pathways could be stopped by using fucoxanthin. Moreover, two PI3K inhibitor agents, LY294002 and wortmannin, when used, could stop the effects of fucoxanthin. This outcome suggested the mechanism by which fucoxanthin exerts its neuroprotective effects could be activation of PI3K/Akt cascade concurrently with stopping ERK pathway. Akt activation by fucoxanthin could also modulate NF-κB concerning reducing oxidative stress [139]. It also decreased apoptosis and oxidative stress in SH-SY5Y cells through activating a pro-survival PI3K/Akt pathway and suppressing the pro-apoptotic ERK pathway and preventing H2O2 stimulated apoptosis [140].

Scopolamine [141], and Aβ oligomer [141] can contribute to cognitive impairments in mice. Fucoxanthin, by inhibiting acetylcholinesterase (AChE) activity, regulation of choline acetyltransferase (ChAT) activity, and increasing BDNF expression has a protective role in these disorders. Nrf2/ARE and Nrf2-autophagy pathways-dependent neuroprotective mechanism is involved in fucoxanthin mediated traumatic brain injury amelioration [142]. Fucoxanthin has also shown promising results against human inflammation-related diseases through employing PI3K/Akt/CREB/peroxisome proliferator-activated receptor-gamma coactivator α, and Nrf2/ARE pathways [127,130]. In a recent study by Sun et al., fucoxanthin inhibited inflammation-related *Lachnospiraceae* and *Erysipelotrichaceae* while increased the *Lactobacillus*/*Lactococcus*, *Bifidobacterium*, and some butyrate-producing bacteria [143]. Guo et al. showed the critical role of fucoxanthin in modulating the ratio of *Firmicutes*/*Bacteroidetes* and the abundance of *Akkermansia*, thereby, it could be an auspicious microbiota-targeted functional food [144]. It has also shown promising interaction with intestinal *Escherichia coli* and *lactobacilli* towards the growth inhibition of pathogenic bacteria [145]. So, fucoxanthin develops the GM and modulates the neuronal inflammatory/oxidative/apoptotic pathways, thereby attenuates the gut-brain axis towards neuroprotective responses.

Most known carotenoids generated by marine fungi are commercially available astaxanthin and β-carotene [117], which the former is a xanthophyll carotenoid with the most potent antioxidant [146]. Extraction of astaxanthin is done due to its lipophilic nature by solvents, acids, microwave coupled, and enzyme aided methods [147]. In a red basidiomycetous yeast named *Phaffia rhodozyma*, astaxanthin has been extracted from the cytoplasmic membrane [88]. Cardinal microorganisms with the ability to synthesize astaxanthin are microalgae *Chlorella zofingiensis*, *Chlorococcum* spp., red yeast *Phaffia rhodozyma*, and the marine *Agrobacterium aurantiacum* [148]. Consequently, algae, yeast, and crustacean produce astaxanthin as a byproduct. Brain higher susceptibility to oxidative stress is due to its excessive metabolism, the existence of already oxidized molecules like catecholamine neurotransmitters, and polyunsaturated fatty acids which are in the structure of the cell membrane. Further exploration revealed other biological activities and health benefits could be obtained from this compound [149,150]. The anticancer [151], anti-obesity/triglyceride/cholesterol, cardioprotective, [152,153], hepatoprotective [154], and anti-diabetic [155] effects of astaxanthin are also reported [149].

The neuroprotective property of astaxanthin has been recently noticed passing through its anti-inflammatory, anti-apoptotic, and antioxidant effects besides maintenance of neural plasticity [156,157,158]. This collection of features, candidate astaxanthin as a therapeutic agent in neurodegenerative diseases. Although, because of its anti-inflammatory and antioxidative behavior, it has been investigated in cardiovascular health, metabolic syndrome, management of gastric ulcers, and malignancy, all of which share a common characteristic of inflammation and oxidative stress [159]. In building to cell membrane against oxidative stress, which is being exacerbated by aging, astaxanthin plays an important part by applying associated antioxidant agents like SOD, heme oxygenase-1 (HO-1), and GSH. Its unique structure of ketone-bearing ionone rings stabilizes radicals synergistically with the polyene backbone and improves antioxidative ability [160]. It was demonstrated that ROS could be depleted in vitro by using astaxanthin [161]. Suppression of microglial activation and as a result, lowering the production of cytotoxic compounds, is another function of astaxanthin [118]. Released NO from microglia during inflammation, reacts with superoxide leading to the formation of peroxynitrite that is a ROS, and destructing proteins/lipids/DNA is the final consequence [162]. It was demonstrated that the expression and release of IL-6, COX-2 was down-regulated by astaxanthin in an LPS-induced activated microglia, in vitro [163]. The host immune system can be boosted by the administration of astaxanthin. The result was the stimulation of T cells which results in the production of IFN-γ, B cells to end up to secretion of IgA, and natural killer cells, as well as generation of IFN-c and IL-6 [164].

In a recent study by Wu et al., astaxanthin made a shift in the GM towards the suppression of intestinal inflammation/oxidative stress pathways by inhibiting colon NLRP3 inflammasome activation [165]. Overall, on one hand, astaxanthin potentially attenuates neuronal dysregulated pathways. Besides, it modulated GM towards the modulation of intestinal inflammation/oxidative stress/apoptosis. Meanwhile, the expression of inflammatory factors including TNF-α and INF-γ decreased, while the expression of IL-10 increased. In another study, by regulating cecal microbiota, astaxanthin meaningfully decreased the expression of MyD88, TLR4, and p-p65, while increasing the p65 expression [166]. So, considering the simultaneous modulatory role of astaxanthin on the gut and brain, it introduces astaxanthin as a great candidate in regulating the gut-brain axis towards neuroprotection.

Lycopene is another carotenoid through which the gut and brain potentiate their beneficial relationship. It exists in vegetables and red fruits, such as red pepper, papaya, tomato, watermelon, and also in algae. Following 5 days of administration of diets on lycopene, the related plasma amount of lycopene is significantly increased towards distribution in blood and brain. Lycopene seems to be responsible for a gut-brain regulation then is particularly discharged undigested from the human body [167]. Studies indicated that the GM balance has a vital contribution to the occurrence/development of colitis, which affects brain function by the gut physiology regulation [3,168]. As a carotenoid, it has shown a critical role in colitis and the associated behavioral disorders. It is shown that 40 days of lycopene therapy (50 mg/kg body weight/day) in male mice prevented gut damages and inflammatory responses induced by dextran sulfate sodium (DSS). Lycopene attenuated DSS-induced dysfunctions of anxiety-like behavioral and depression by suppressing synaptic damages, inhibiting neuroinflammation, and increasing the expressions of neurotrophic factor and postsynaptic-density protein. Additionally, lycopene resulted in GM remodeling of colitis mice by reducing the relative abundance of *Proteobacteria* and elevating the relative abundance of *Lactobacillus* and *Bifidobacterium*. It has also stimulated the SCFAs production and suppressed the LPS permeability in colitis mice [169]. Recent results demonstrated that GM germinates SCFAs in *Bacteroides* and some others. Among those, *Prevotella*, *Ruminococcus*, *Bacteroides*, *Clostridium*, and *Streptococcus* produce acetic acid, *Bacteroides*, *Coprococcus,* and *Ruminococcus*, germinate propionic acid, and *Lachnospiraceae Bifidobacterium*, and *Coprococcus* harvest butyric acid [170]. The cell wall of Gram-negative bacteria (e.g., *Proteobacteria* and *Bacteroides*) is primarily composed of LPS, which is an activator antigen of TLR4 in triggering inflammatory responses [169]. The dietary supply of *Bifidobacteria* and *Lactobacilli* has also shown promising effects on behavioral disorders like depression [168]. Growing reports have indicated that the destruction of the gut barrier would also result in a gut permeability increment. Subsequently, the generation of some pathogenic compounds (e.g., LPS) is followed by passing through intestinal epithelium and the blood entrance, to trigger neuroinflammation [171]. Reports have revealed that microglia activation and pro-inflammatory cytokines (e.g., IL-1 and TNF-α) could dysregulate synaptic plasticity towards emotional disorders and depression-like behaviors [127]. Thus, lycopene attenuates DSS-induced behavioral disorders and colitis through making a balance in the microbes-gut-brain axis, thereby reducing brain microglia activation and gut/brain inflammatory cytokines [172]. In agreement with recent studies, a low level of BDNF may increase stress susceptibility to affect the structure of forebrain neurons. Therefore, it has been an auspicious marker of a positive antidepressant response of related drugs [3]. Moreover, lycopene has also protected the brain against behavioral disorders by modulating the brain expression of BDNF, as well as associated synaptic plasticity. Bidirectionally, it is controlling the DSS-induced change of GM metabolites including SCFAs and LPS. Thus, lycopene is reshaping the GM of DSS-induced colitis mice, indicating that gut-brain axis balance is an underlying mechanism [167]. Overall, the studies illustrated that lycopene could ameliorate DSS-induced colitis, as well as associated anxiety-like symptoms and depression. It could be clarified by the helpful effects of lycopene on gut barrier integrity as well as its modulatory roles on GM and associated production of metabolites [172].

As previously mentioned, lycopene has shown beneficial effects in CNS. Its activity in reducing oxidative stress and tert-butyl hydroperoxide-induced cell apoptosis promised using lycopene in AD. Lycopene has a variety of activities including raising GSH/GSSG enzyme levels, maintenance of mitochondrial membrane potential, depletion of ROS [173], reduction of inflammatory cytokine levels, and downregulation of TLR4 and NF-κB p65 mRNA and protein expressions [174]. Besides, the effects of lycopene on neurological recovery and anti-inflammation activities were also investigated on rat models for the management of spinal cord ischemia/reperfusion injury [175]. During a clinical trial, lycopene increased the relative abundance of GM profile, such as *Bifidobacterium adolescentis* and *Bifidobacterium longum* [176]. So, according to the critical role of lycopene in modulating GM and associated mediators of the gut-brain axis, it would be a promising agent towards neuroprotection.

### 5.2. Polysaccharides: Fucoidan, Chitosan, Alginate, and Laminarin

Other anti-inflammatory molecules that could be obtained from algal are polysaccharides, including chitosan, laminarin, fucoidan, and alginate [177]. Attenuation of NF-κB and ERK/MAPK/Akt pathways in BV2 microglia induced by LPS resulted in depressed anti-inflammatory responses by a sulfated polysaccharide named fucoidan [178]. In an in vivo study by Shang et al., fucoidan attenuated the GM in mice by increasing the abundance of *Ruminococcaceae* and *Lactobacillus* [179]. In another study, it alleviated colonic inflammation and GM dysbiosis by inhibiting IL-1β, IL-6, and TNF-α, while increasing IL-10 [180]. In a more recent study, feeding fucoidan from *Okinawa mozuk* altered the intestinal microbiota composition of adult zebrafish, which was characterized by the emergence and predominance of multiple bacterial affiliated with *Comamonadaceae* and *Rhizobiaceae*. They also found decreased expression of intestinal IL-1β [181]. Shi *et al*., also showed that fucoidan from *Acaudina molpadioides* increased the abundance of short SCFAs producer *Coprococcus*, *Butyricicoccus,* and *Rikenella*, through which mitigated intestinal mucosal injury [182]. In a recent double-blinded, placebo-controlled study, combination therapy of fucoidan increased fecal SCFAs, *Pseudocatenulatum*, *Bifidobacterium*, *Bacteroides intestinalis*, *Eubacterium siraeum*, while decreased *Prevotella copri* [183]. Additional studies have also focused on the potential of fucoidan in modulating GM by targeting antioxidative mediators (e.g., Nrf2, and GSH/GSSG) [184]. Sulfated oligosaccharides of gran algae *Ulva lactuca* and *Enteromorpha prolifera* decreased pro-inflammatory agents, reduced p53 and fork-head box protein O1 (FoxO1) genes, and caused overexpression of Sirt1 gene in SAMP8 mice [185]. Additionally, it upregulated the expression of BDNF through the ERK/CREB/tropomyosin-related kinase receptor B (TrkB) pathway and antioxidant enzymes such as HO-1, NAD(P)H quinine oxidoreductase-1 (NQO-1), and glutamate-cysteine ligase catalytic subunit (GCLC) via Akt [186].

The second most abundant polysaccharide is chitin after cellulose [187]. The deacetylated derivative, chitosan, has shown several therapeutic applications, including antimicrobial/antifungal [188,189], antioxidant [190], and antitumor [191] effects. Many studies showed the neuroprotective, antioxidative, and anti-inflammatory properties of chitosan. Deacetylation degree and chain size are two determinants of these properties [192]. In a study by He et al., chito-oligosaccharides were used as a protectant against oxidative stress caused by H_2_O_2_, and the lowest concentration with the highest efficacy was reported as 0.02 mg/mL. It was also demonstrated that carboxymethylated chitosan had a satisfactory effect in protecting Schwann cells against hydrogen peroxide-induced damage through a mitochondrial-dependent pathway [193]. In most recent reports, chitosan potentially modulated GM [194] by suppressing *Helicobacter*, promoting *Akkermansia* [195], and decreasing serum levels of IL-6 and IL-1β [196]. Yu et al., also showed that chitosan supplementation improved the *Prevotella* in the cecal of pigs [197]. Another polysaccharide that has been extracted from algae is seleno-polymannuronate was prepared from alginate-derived polymannuronate. Alginate is the major structural polysaccharide of brown macroalgae, which is widely used as food additives and functional food ingredients owing to the desired physicochemical properties and well-recognized beneficial effects on gut ecology [198]. *Bacteroides ovatus* is responsible for the fermentation of alginate in the gut. The fermentation outcome of alginate by gut microorganisms are SCFAs, which acts as an energy provider for intestinal and immune cells [199]. Treatment with alginate reduced IL-1β and CD11c inflammatory markers, and improved the growth of GM *Lactobacillus gasseri*, *Lactobacillus reuteri*, and *Akkermansia muciniphila* in mice [200]. Such beneficial roles of alginate on GM were also reported in other studies [201]. Additionally, alginate-derived oligosaccharide downregulated pro-inflammatory agent’s expression and enzymes in LPS/Aβ-induced BV2 microglia. This oligosaccharide also decreased the expression of TLR4 and NF-κB [202]. From the mechanistic point, alginate has targeted oxidative pathways (e.g., ROS) and neuroinflammation (e.g., TLR4) to protect neuronal damages. So, regulation of GM and neuronal dysregulated pathways by alginate pave the road in attaining a promising gut-brain axis [203].

Other polysaccharides which could be fermented by GM include Laminarin [204]. Brown seaweeds, such as *Laminariaceae* as the source and *Laminaria*, *Saccharina*, or *Eisenia* species as secondary [205] contain a linear polysaccharide named laminarin. Studies showed its neuroprotective properties in cerebral ischemic events through gliosis reduction and controlling pro-inflammatory microglia. In an in vivo study, laminarin meaningfully decreased GM *Firmicutes* while increased *Bacteroidetes* [206]. On the other hand, laminarin shows potential neuroprotective effects by attenuation of inflammation and oxidative stress. In this regard, laminarin reduced IL-1β, TNF-α, while increased SOD, IL-4, and IL-13 in aged gerbils [207]. So, by targeting GM and triggering anti-inflammatory and antioxidant mediators, laminarin could pave the road in the modulation of the gut-brain axis towards neuroprotection.

Besides, carrageenan derived from red algae is a high molecular weight biopolymer molecule, involves linear sulfated galactans [205]. κ-carrageenan extracted from *Hypnea musciformis* red algae showed the neuroprotective property in neurotoxicity caused via 6-hydroxydopamine on SH-SY5Y cells by regulating mitochondria transmembrane potential and decreasing activity of caspase-3 [208]. Porphyrin from *Porphyra yezoensis* suppressed NO generation in LPS-stimulated RAW264.7 cells by downregulation of iNOS expression [209]. Besides, sulfated polysaccharide fractions from *Porphyra haitanesis* had antioxidative properties and avoided lipid peroxidation in rat liver microsome [131].

In general, marine-derived polysaccharides either modulate GM and suppressed neuronal dysregulations towards the attenuation of the gut-brain axis and developing neuroprotection.

### 5.3. Macrolactins/Anthraquinones: Macrolactin A

Another example of marine natural products is macrolactins. Those are polyene cyclic backbone consisting of 24-membered ring lactones with alterations such as the binding of glucose β-pyranoside. Around 32 macrolactins are discovered so far including, 7-*O*-succinyl macrolactin A, macrolactins A–Z, 7-*O*-succinyl macrolactin F, 7-*O*-malonyl macrolactin A, and three ether-containing macrolactin A. Their sources are marine sediment, and soil isolates. Macrolactin A has exhibited antibacterial effects against *S. aureus* and *B. subtilis* at a concentration of 5 and 20 µg/disc, respectively, with the ability to inhibit B16-F10 murine melanoma cancer cells in vitro, mammalian *Herpes simplex* viruses, and protection of lymphoblast cells in opposition to HIV by inhibiting virus replication [210]. A Gram-positive bacterium which is a habitant of the deep-sea sediments is another source of macrolactin A. This compound has antibacterial, antiviral, and cytotoxic activities. It shows potent inhibition of mammalian herpes simplex virus (type I and II), protection of T cells against HIV replication, and suppression of B16-F10 murine melanoma cells in in vitro assays [211]. Marine microorganism-derived macrolactins inhibit inflammatory mediators through modulating HO-1/Nrf2 and suppressing TLR4 [212]. Yan *et al*. showed that treatment with macrolactins inhibited the mRNA expressions of iNOS, IL-1β, and IL-6, in vitro [213]. Those all could introduce macrolactins as promising modulators of the gut-brain axis.

In addition to macrolactin A, anthraquinones, like carotenoids, have antioxidative properties [88,214] and this property is owed to the quinoid structure that makes them capable to take part in reduction-oxidation reactions [215]. The antioxidant/pro-oxidant [216], antimicrobial/antiviral/antiparasitic [217], immunomodulatory [218], diuretic/laxative [109], vasorelaxant [219], lipid/glucose-lowering [220], and estrogenic activities [221] of anthraquinones extracted from marine sources are also reported [222]. These classes of marine drugs have shown potential neuroprotective responses by suppressing inflammatory/apoptotic/oxidative pathways [223]. Referring to the gut-brain axis, anthraquinones meaningfully promoted the dominant growth of *Akkermansia muciniphila* and suppressed the growth of *Clostridium tyrobutyricum* and *Clostridium butyricum* as well as butyrate-producing bacteria, thereby decreasing butyrate levels [224].

So, considering the neuroprotective role of macrolactins/anthraquinones as well as their modulatory effects on GM could make their acceptable future on the gut-brain axis towards neuroprotection.

### 5.4. Diterpenes/Sesquiterpenes: Lobocrasol, Excavatolide B, Crassumol E, and Zonarol

Soft corals are reservoirs of other marine-derived structurally named diterpenes as secondary metabolites with antioxidative properties. They participate in immune responses via regulation of the NF-κB signaling pathway at different levels [225,226]. A diterpene from cultured Taiwan gorgonian *Briareum excavatum*, called Excavatolide B, could suppress mRNA expression of COX-2 and iNOS in LPS-induced macrophage-activated mouse models through anti-inflammatory effects [227]. NF-κB activation is induced by TNF-α at sites of inflammation in various diseases, and thereby strong inhibition of NF-κB activation was observed by Lobocrasols A and B and other cembranoid diterpenes (crassumol E and (1*R*,4*R*,2*E*,7*E*,11*E*)-cembra-2,7,11-trien4-ol) that were obtained from methanolic extract of the Vietnamese soft coral *Lobophytum crassum* [228]. Novel diterpenoid compounds from *L. crissum* also showed anti-inflammatory effects by inhibiting COX-2 and iNOS [229]. Subsequently, as a sesquitepene, zonarol is derived from *Dictyopteris undulata* marine source with potential antioxidant effects (increasing NQO-1, HO-1, and PRDX4) to modulate the gut-brain axis [139,230]. These effects could introduce diterpenes/sesquiterpenes as promising therapeutic agents in modulating gut-brain axis and neuronal dysfunction.

### 5.5. Phytosterols: Fucosterol and Solomonsterol A

Steroids with strong anti-inflammatory activity and immune response suppression could be found in marine sponges. Receptors of farnesoid X and pregnane X were modulated by steroidal molecules [231]. Agonists of pregnane X receptors demonstrated promising effects in lowering intestinal inflammation and NF-κB activity [232]. Fucosterol, as a phytosterol, with a diverse range of activities including, antioxidant, anti-inflammatory, anticholinesterase, neuroprotection is extracted from brown algae. Besides, it can inhibit acetyl- and butyryl-cholinesterase (AChE and BChE) and β-secretase (an enzyme involved in Aβ production, which is a key player in AD) and diminish Aβ-induced neuronal death [233]. Recent studies have proved the potential of fucosterol to be used in neurodegenerative diseases. Although its precise mechanism is unknown. Due to its cholesterol-like structure, targets like intracellular proteins are reachable for this compound, and it can cross the cell membrane. Literature evidence showed that the target proteins of fucosterol are involved in inflammatory pathways like TNF, hypoxia-inducible factor 1-alpha (HIF-1), NF-κB, and vascular endothelial growth factor (VEGF) signaling. NF-κB as a transcription factor takes part in inflammatory and immune responses and so has a major role in neurodegeneration diseases. In research, it was showed that fucosterol has a weakening effect on inflammation induced by LPS in RAW 264.7 macrophages [234].

Cholesterol homeostasis in the brain is regulated by LXRs which are nuclear receptors with cholesterol-sensing ability. After activation of these receptors, they inhibit the production of pro-inflammatory cytokines such as TNF-α and IL-1β. Besides LXRs affect the expression of genes related to lipid metabolism, they also block inflammatory gene expression initiated by TLR activation [234]. PI3K/Akt pathway in parallel to MAPK signaling pathway modulates growth and survival of the cell. These pathways downstream compounds like CREB, Bcl-2, caspase-9, IKK, and NF-κB also participate in cell survival processes. It has been demonstrated that CREB which is a transcription regulator sensing the upstream signal from PI3K/Akt pathway was targeted by fucosterol. CREB has a pro-survival activity through upregulation of Bcl-2. Cholinergic, dopaminergic, and serotonergic synapses are highly abundant with CREB, ensuing that fucosterol might have a major effect on neuronal growth, survival, and activity. Maturation of evolving neurons, growth, and survival of mature neurons are dependent on the neurotrophin signaling pathway. Therefore, any abnormality in this pathway will cause neurodegeneration. Evidence showed that neurotrophin mimetic agents could be used in the therapy of AD. TrkB as a receptor of BDNF is one of the targets of the fucosterol which confirms its neurotrophin mimetic activity [234].

Aβ1-42-induced cytotoxicity through activating TrkB-mediated ERK1/2 signaling in primary hippocampal neurons could be inhibited by fucosterol, based on a work done by Oh and colleagues in an in vitro assay. Translation of in vitro results was done by showing the effects of fucosterol on the attenuation of Aβ1-42-stimulated mental impairment in aging rats [15]. Molecular docking and binding energy calculation in an in silico analysis confirmed strong binding affinity of fucosterol to TrkB which could be assumed as extra evidence of BDNF-mimetic activity of fucosterol.

Antioxidant activity of the fucosterol was proved by its effect of raising the levels of antioxidant enzymes such as glutathione peroxidase (GPX1), SOD, CAT, and HO-1 via Nrf2 activation and in silico data on TrkB binding [235]. Synaptic plasticity can be regulated by Ca^2+^ signaling at glutamatergic synapses. Synaptic proteins, such as GluN2A and AChE which have an affinity to fucosterol are associated with Ca^2+^ signaling, thus influences synaptic plasticity that affects memory and cognition. Based on the above information, fucosterol as a GluN2A NMDA receptor agonist could be used to improve AD patients [234].

The cholinergic deficit has a role in the pathology of AD. AChE inhibitor, by increasing availability of the acetylcholine, can compensate this deficiency, so fucosterol with AChE inhibitory property, (confirmed by molecular docking findings) can be used efficiently in the treatment of AD patients [236]. TLRs are key role-players of innate immunity by detecting pathogen-associated molecular patterns. Pathogenesis of chronic diseases including neurodegeneration is associated with the malfunction of TLR signaling. An in silico analysis with TLRs as protein target showed interaction of the fucosterol with both TLR2 and TLR4 which leads to the idea of using it in inflammation-induced neurodegeneration.

Levels of antioxidant activity of SOD, GPx, and CAT were raised in rat models induced by fucosterol [237]. Jung and colleagues showed that fucosterol avoided ROS generation in tert-butyl hydroperoxide (t-BHP)-induced RAW264.7 macrophages [238]. Also, in another research, it was concluded that HepG2 cells were protected against oxidation via fucosterol [239] and its mechanism by which the lung epithelial cells were protected was the elevation of SOD, CAT, and HO-1, and nuclear translocation of Nrf2 [240]. The glycoprotein of *U. pinnatifida* elevated activity of SOD (53.45%) and the activity of xanthine oxidase reduced by 82.05%. Diphlorethohydroxycarmalol and 6,6′-bieckol from *Ishige okamurae* showed antioxidant activity and decreased ROS level in RAW264.7 cells [131].

During the consequent studies, it was observed that inflammation induced by LPS was attenuated via using fucosterol in RAW 264.7 macrophage [112] and alveolar macrophage [241]. LPS- or Aβ-mediated neuroinflammation in stimulated microglial cells also was attenuated by fucosterol. *Ecklonia* spp. produces phlorotannins such as dieckol, phlorofucofuroeckol A [242], phlorofucofuroeckol B 6,6′-bieckol, and 8,8′-bieckol with anti-inflammatory properties via suppression of NF-κB and MAPK pathways [243].

As previously mentioned, some other classes of marine natural products also play critical roles in modulating the gut-brain axis. Of those, neoechinulin B, as an alkaloid extracted from *Eurotium* sp. SF-5989 marine fungi. is in this way through suppressing NF-κB and p38MAPK towards inhibiting neuroinflammation [244].

In general, carotenoids, polysaccharides, phytosterols, terpenoids, phenolic compounds, and alkaloids extracted from marine sources, possess several biological activities and health benefits. Figure 1 displays the chemical structure of the aforementioned candidate marine natural products capable of modulating the gut-brain axis towards neuroprotection.

They have the potential of modulating GM towards neuroprotection, thereby attenuate the gut-brain axis. Table 2 indicates marine-derived compounds, as well as associated sources towards neuroprotective responses.

The bidirectional relationship of GM and brain, as well as the critical role of marine natural products, are provided in Figure 2. As described, marine natural products are modifying the GM towards modulation of critical dysregulated inflammatory/apoptotic/oxidative stress pathways in neurodegenerative diseases, thereby reveal neuroprotection.

## 6. Conclusions

In addition to the role of GM in modulating dysbiosis, its critical role in preventing/treating neurodegenerative diseases (e.g., AD, ASD, ALS, MS, and PD), as well as associated complications is undeniable. Growing evidence implicates the gut-brain axis as a possible key target in the attenuation of neurological disorders.

Marine natural products are multi-target agents in a simultaneous modulation of intestinal and supra-intestinal disorders. Accordingly, carotenoids, polysaccharides, phytosterols, terpenoids, phenolic compounds, and alkaloids either by indirect modulation of GM or by direct suppression of neuroinflammation/apoptosis/oxidative stress, could be considered a potential/efficient strategies in combating neurodegenerative diseases. To do such, marine natural products potentially reduce the relative abundance of harmful GM, while increasing beneficial GM towards modulating the inflammatory mediators (e.g., NF-κB, TNF-α, ILs, COX-2, and TLRs), apoptosis (e.g., caspase, Bax/Bcl-2), and oxidative stress (e.g., ROS, Nrf2, HO-1, and AREs) in the gut. These compounds also regulate associated critical pathways in the gut, including PI3K/Akt/mTOR, MAMPs, BDNF, and ERK/CREB/MAPK. Considering the bidirectional relationship between gut and brain, modulation of these mediators/signaling mediators in the gut would result in their regulation in the brain towards neuroprotection.

A further area of research should focus on pre-clinical studies to reveal the precise molecular communication of the gut-brain axis, followed by well-controlled clinical trials. Such studies will help to investigate the more potential marine-derived natural products in the prevention, management, and treatment of the gut-brain axis towards neuroprotection.

## Figures and Tables

**Figure 1 marinedrugs-19-00165-f001:**
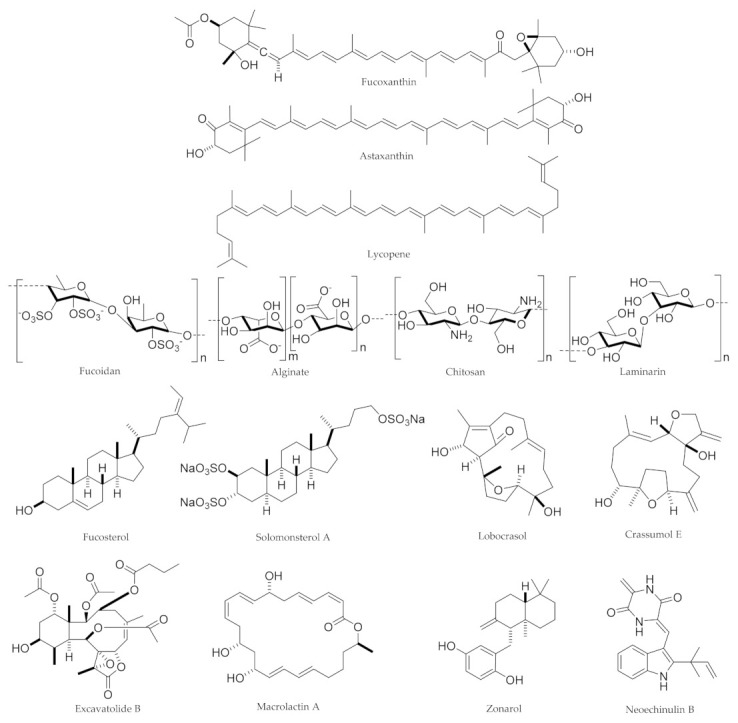
Chemical structure of candidate marine natural products.

**Figure 2 marinedrugs-19-00165-f002:**
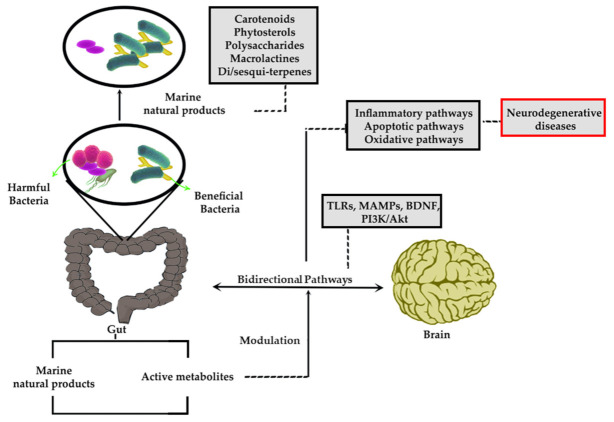
Marine natural products and gut-brain axis. BDNF: brain-derived neurotrophic factor, MAMPs: microbes associated molecular patterns, PI3K: phosphoinositide 3-kinase, TLRs: tool-like receptors.

**Table 1 marinedrugs-19-00165-t001:** Gut-brain axis in neurodegenerative diseases and associated outcomes.

Neurological Disorder	Changes of Microbiota	Effects/Outcomes	Reference
AD	*Bacteroides vulgatus, Bacteroides fragilis, Eggerthella lenta,* *Odoribacter splanchnicus, Butyrivibrio hungatei, Butyrivibrio proteoclasticus, Eubacterium eligens, Eubacterium hallii, Eubacterium rectale, Clostridium* sp.*, Roseburia hominis, Bifidobacterium bifidum, Faecalibacterium prausnitzii*	↑TLRs, ↑NF-κB, ↑IL-1β, ↑IL-18, ↑ Aβ, ↑caspase-1, CXCL2, ↑bacterial LPS,	[51,52,53,54]
PD	*Enterobacteriaceae, Prevotellaceae, Verrucomicrobiaceae, Lactobacillus, Porphyromonas, Parabacteroides, Mucispirillum, Bacteroides fragilis*	↑TLR4, ↑IL-1β, ↑IL-2, ↑IL-4, ↑IL-6, ↑IL-13, ↑IL-18, ↑IFN-γ, ↑TNF-α	[3]
ASD	*Bifidobacteraceae, Veillonellaceae, Lactobacillaceae, Bifidobacterium, Megasphaera, Mitsuokella, Rumnicoccus, Lachnoclostridium, Clostridium, Sutterella, Desulfovibrio, Lactobacillus, Eubacterium, Prevotella*	↑mTOR, ↑TNF-α, ↑IL-4, ↑IL-5, ↑IL-6, ↑IL-8, ↑valeric acid, ↑intestinal serotonin, ↓IL-10, ↓TGF-β, ↓fecal acetic acid and butyrate, ↓cerebral 5-HT	[11,81,84,85,86,87]
Depression	*Bifidobacterium, Alistipes, Prevotella, Parabacteroides, Lachnospiraceae, Anaerostipes, Oscillibacter, Faecalibacterium, Ruminococcus, Clostridium, Megamonas, Streptococcus, Klebsiella, Phascolarctobacterium*	↓GABA, ↓dopamine, ↓5-HT, ↓BDNF, ↓IL-10	[63,64,65]
ALS	*Ruminococcaceae, Bacteroidetes, Enterobacteria, Escherichia coli,* *Butyrivibrio fibrisolvens,* *Bacterioidetes, Oscillobacte,* *Firmicutes, Anaerostipes, Lachnospiraceae*	Dysregulated levels of NO, GABA, LPS, AMPA/NMDA, and oxidative pathways	[74,75,76]
MS	*Acinetobacteria, Bacteroidetes, Desulfovibrionaceae, Firmicutes, Proteobacteria, Verrucomicrobia,* and associated genus	↓5-HT, ↓dopamine, dysregulated GABA, ↑IFN-*γ*, ↑MCP-1, ↑MIP-1*α*, ↑MIP-1*β*, ↑IL-6	[77,80,81,82]
ASD	*Bifidobacteraceae, Veillonellaceae, Lactobacillaceae, Bifidobacterium, Megasphaera, Mitsuokella, Rumnicoccus, Lachnoclostridium, Clostridium, Sutterella, Desulfovibrio, Lactobacillus, Eubacterium, Prevotella*	↑mTOR, ↑TNF-α, ↑IL-4, ↑IL-5, ↑IL-6, ↑IL-8, ↑valeric acid, ↑intestinal serotonin, ↓IL-10, ↓TGF-β, ↓fecal acetic acid and butyrate, ↓cerebral 5-HT	[11,81,84,85,86,87]

Aβ: amyloid-beta, AD: Alzheimer’s disease, ALS: amyotrophic lateral sclerosis, ASD: autism spectrum disorder, BDNF: brain-derived neurotrophic factor, CXCL2: chemokine (C-X-C motif) ligand 2, GABA: γ-aminobutyric acid, IFN-γ: interferon-gamma, IL: interleukin, LPS: lipopolysaccharide, MCP: monocyte chemoattractant protein, MIP: macrophage inflammatory protein, MS: multiple sclerosis, mTOR: mammalian target of rapamycin, NF-κB: nuclear factor-κB, NLRP3: NLR family pyrin domain containing 3, NMDA: N-methyl-D-aspartate, PD: Parkinson’s disease, TGF-β: transforming growth factor-beta, TLR: toll-like receptor, TNF-α: tumor necrosis factor-alpha, 5-HT: serotonin.

**Table 2 marinedrugs-19-00165-t002:** Marine-derived compounds, sources, and associated neuroprotective responses.

Marine Class	Compound	Major Source	Effect/Outcome	Reference
Carotenoid	Fucoxanthin	*Sargassum siliquastrum, Hijikia fusiformis,* *Undaria pinnatifida, Laminaria japonica, Alaria crassifolia, Cladosiphon okamuranus*	↑BDNF, ↑SOD, ↓ROS, ↓MDA, ↓cleaved caspase-3, ↑Bcl-2/Bax ratio	[117,127]
			↓ROS, ↑Beclin-1 (Atg6), ↑LC3 (Atg8) and ↓p62, ↓cleaved caspase-3, ↑HO-1, ↑NQO-1↑Nrf2	[139,245]
	Astaxanthin	*Hematococcus pluvialis*, *Chlorella zofingiensis*, *Chlorococcum sp., Phaffia rhodozyma*, *Agrobacterium aurantiacum*	↓Bax/Bcl-2 ratio, ↓caspase-3, ↓Ca^2+^ influx, ↓ROS, ↓MDA, ↓LPO, ↓IL-1β, ↓TNF-α, ↓OS, ↓ NF-κB, ↓IL-1β, ↓ICAMs1	[149]
	Lycopene	*Haloarchaea* belonging to *the Haloferacaceae family*	↑GSH/GSSG, ↑BDNF, ↓TNF-α, ↓NF-κB, ↓ILs, ↓TLR4	[167,174]
Phytosterol	Fucosterol	*Anthophycus longifolius, Chondria dasyphylla, Ecklonia* *stolonifera, Undaria pinnatifida, Hizikia fusiformis*	↑TrkB-mediated ERK1/2, ↓GRP78 ↑BDNF, ↑Ngb mRNA ↓APP, ↓Aβ levels	[15,246,247]
	Solomonsterol A	*Theonella swinhoei*	↓Arthritic score in anti-type II collagen, antibody-induced arthritis mice model	[248]
Polysaccharide	Sulfated polysaccharide	*Porphyra haitanensis, Ecklonia cava, Laminaria japonica,* *Cladosiphon okamuranus*	↓IgE level in tropomyosin-induced mouse allergy model	[249]
	Fucoidan		↑p-PKC, ↓OS ↓caspases-9/3, ↓ROS, ↓LC3-II, ↑SOD, ↑GPx, ↓MDA,↑Bcl-2/Bax ratio, ↓cytochrome C, ↑livin and XIAP; ↑GSH, ↓Bax	[250,251]
	Chitosan	Species of crustaceous and cephalopods	Modulating mitochondrial-dependent pathway	[189,190,193]
	Laminarin	Brown seaweeds such as *Laminariaceae,* and *Laminaria*, *Saccharina*, or *Eisenia*	↓Pro-inflammatory microglia	[204,205]
	Alginate	Microalgae	↓TLR4, ↓NF-κB, ↓ROS	[198,199,202]
Diterpene	Excavatolide B	*Briareum excavatum*	↓iNOS, ↓COX-2	[227]
	Crassumol E	Soft coral *Lobophytum crassum*	↓NF-κB, ↓TNF-α	[227]
	Lobocrasol	Soft coral *Lobophytum crassum*	↓NF-κB	[228]
Hydroquinone sesquiterpene	Zonarol	*Dictyopteris* *undulata*	↑NQO-1, ↑HO-1, ↑PRDX4	[139,230]
Macrolactin	Macrolactin A	*Bacillus subtilis*, Marine sediment, and soil isolates	↑Nrf2, ↑HO-1, ↓NF-κB, ↓TLR4, ↓IL-6, ↓iNOS	[212,213]
Alkaloid	Neoechinulin B	*Eurotium* sp. SF-5989	↓NF-κB, ↓p38 MAPK	[244]

APP: amyloid-beta precursor protein, BDNF: brain-derived neurotrophic factor, ERK: extracellular-regulated kinase, GPx: glutathione peroxidase, GSH: glutathione, HO-1: heme oxygenase-1, ICAMs-1: soluble intercellular adhesion molecule 1, iNOS: inducible nitric oxide synthase, LC3: light chain 3, MAPK: mitogen-activated protein kinase, MDA: malondialdehyde, Ngb: neuroglobin, NO: nitric oxide, NF-κB: nuclear factor-κB, NQO-1: NAD(P)H Quinone Dehydrogenase 1, Nrf2: nuclear factor erythroid 2–related factor 2, OS: oxidative stress, PKC: protein kinase C, ROS: reactive oxygen species, SOD: superoxide dismutase, TLR: toll-like receptor, TNF-α: tumor necrosis factor-alpha, TrkB: tropomyosin-related kinase receptor B.

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
