# Peer review of "Marine Natural Products: Promising Candidates in the Modulation of Gut-Brain Axis towards Neuroprotection"

_marinedrugs, 2021, doi:10.3390/md19030165_

Round 1

Reviewer 1 Report

The MS by Fakhri et al. describes the importance of gut microbiota in neurological diseases, discussing the possibilities to use marine natural products as promising candidates with a neuroprotective role in the gut-brain axis.

The content is appropriate for the Journal Marine Drugs, and the work is interesting because it represents the first review highlighting the potential roles of marine-derived natural products in modulating the gut-brain axis towards neuroprotection. However, the manuscript cannot be accepted in the current version but needs major revisions.

I have comments that authors have to address to improve the MS:

  • Table 1, inserted at the end of the manuscript, is not included in the text part. Please include Table 1 in the text, inserting it after section 3.
  • Table 2, inserted after Table 1, is not included in the text part. Why are many molecules from Table 2 not discussed in the section 5?
  • Section 5: this key part does not highlight the molecules from marine sources. Please emphasize these molecules and insert them in the table 2, if applicable.
  • For the aim of the journal, it is worth including the chemical structures of marine natural products discussed in table 2 and section 5

Minor points:

-Lines 58 and 393: replace miss folding with misfolding

-Line 60: delete coma

- Please check along the text the names of microorganisms and families of microorganisms in Italics

-Table 2: AST stands for?

Author Response

Overall comment: The MS by Fakhri et al. describes the importance of gut microbiota in neurological diseases, discussing the possibilities to use marine natural products as promising candidates with a neuroprotective role in the gut-brain axis. The content is appropriate for the Journal Marine Drugs, and the work is interesting because it represents the first review highlighting the potential roles of marine-derived natural products in modulating the gut-brain axis towards neuroprotection. However, the manuscript cannot be accepted in the current version but needs major revisions. I have comments that authors have to address to improve the MS.

Response: We greatly express our sincere thanks to the respected reviewer for the careful review and positive consideration of our manuscript towards publication. The “point-by-point” and itemized modification has been done in light of your comments.

 Comment 1: Table 1, inserted at the end of the manuscript, is not included in the text part. Please include Table 1 in the text, inserting it after section 3.

Response 1: Thank you. Table 1 is now included in the text part after section 3.

Comment 2: Table 2, inserted after Table 1, is not included in the text part. Why are many molecules from Table 2 not discussed in the section 5?

Response 2: Thanks again. Table 2 is now inserted in the text part. Besides, we have completed the explanation on the molecules of Table 2 in section 5; however, some molecules are more explained in the text, owing to their importance and more related studies (e.g., fucoxanthin, astaxanthin, lycopene, etc).

Comment 3: Section 5: this key part does not highlight the molecules from marine sources. Please emphasize these molecules and insert them in the table 2, if applicable.

Response 3: This is a critical comment. We have now highlighted the molecules from marine sources in the tile of each sub-section 5 and abstract then included in Table 2. Additional studies have also been added regarding a more highlighting of marine natural molecules.

Comment 4: For the aim of the journal, it is worth including the chemical structures of marine natural products discussed in table 2 and section 5.

Response 4: This is a critical comment. The chemical structures of candidate marine natural products discussed in table 2 and section 5 are provided in Figure 1.

Comment 5: Minor points:

-Lines 58 and 393: replace miss folding with misfolding

-Line 60: delete coma

Response 5: Thanks for pointing out the issues. They are corrected.

 Comment 6: Please check along the text the names of microorganisms and families of microorganisms in Italics.

Response 6: This is a critical point. The names of microorganisms and associated families are now in italic.

Comment 7: Table 2: AST stands for?

Response 7: Thanks. This is an important point. AST stands for astaxanthin which is now replaced with the full name.

Reviewer 2 Report

The manuscript was well-written, and the results are sound. There are some concerns that should be addressed.

  1. Introduction lacks hypothesis and introduction section can be elaborated.
  2. The fig. 1 should be in the context of article not in conclusion.
  3. Please check for English language carefully.
  4. the form of abstract should be checked again.
  5. The Abstract is unclear and deserves improvement, especially the second half. An unclear and non-appellative Abstract does not attract readers to the entire paper. 
  6. It is not clear why  Marine natural products have modulation of gut-brain axis towards neuroprotection?

Author Response

Overall comment: The manuscript was well-written, and the results are sound. There are some concerns that should be addressed.

Response: We greatly express our sincere thanks to the respected reviewer for the careful review and positive consideration of our manuscript towards publication. The “point-by-point” and itemized modification has been done in light of your comments.

 Comment 1: Introduction lacks hypothesis and introduction section can be elaborated.

Response 1: Thanks. The introduction is now more elaborated and the hypothesis provided.

Comment 2: The fig. 1 should be in the context of article not in conclusion.

Response 2: Thank you for the comment. Figure 1 (newly figure 2) is now in the context of the article before the conclusion.

Comment 3: Please check for English language carefully.

Response 3: We have checked the entire manuscript and did our best to improve the language and correct linguistic problems and mistakes.

Comment 4: The form of abstract should be checked again.

Response 4: Thank you. Substantial changes have been made to improve the abstract.

Comment 5: The Abstract is unclear and deserves improvement, especially the second half. An unclear and non-appellative Abstract does not attract readers to the entire paper. 

Response 5: Thanks for the valuable comment. We have made extensive improvements in the abstract.

Comment 6: It is not clear why Marine natural products have modulation of gut-brain axis towards neuroprotection?

Response 6: Thanks for the comment. Regarding your comment, section 5 is now completed on the effects of marine drugs on the gut-brain axis. Besides, the issue is provided in section 2, especially those parts highlighted in yellow. In addition, a summary of gut-brain axis modulation by marine natural products has been provided in the conclusion section.

Round 2

Reviewer 1 Report

The authors replied to my comments satisfactorily, the manuscript is now improved. However, they did not include all chemical structures in the Figure 1. Solomonsterol A, Excavatolide B, Crassumol E are not inserted. Please include these structures in the Figure 1.

Author Response

Thank you very much for your valuable comment. In Figure 1, the chemical structures of Solomonsterol A, Excavatolide B, and Crassumol E were inserted.

Reviewer 2 Report

The author had corrected the manuscript according to my comments.

Author Response

Thank you very much for your valuable comment.